# The Effect of Virus-Specific Vaccination on Laboratory Infection Markers of Children with Acute Rotavirus-Associated Acute Gastroenteritis

**DOI:** 10.3390/vaccines11030580

**Published:** 2023-03-02

**Authors:** Omer Okuyan, Yusuf Elgormus, Ugurcan Sayili, Seyma Dumur, Ozlem Erkan Isık, Hafize Uzun

**Affiliations:** 1Department of Pediatrics, Medicine Hospital, Istanbul Atlas University, 34408 Istanbul, Turkey; 2Clinic of Pediatrics, Medicine Hospital, 34408 Istanbul, Turkey; 3Department of Public Health, Cerrahpasa Faculty of Medicine, Istanbul University-Cerrahpasa, 34098 Istanbul, Turkey; 4Department of Medical Biochemistry, Faculty of Medicine, Istanbul Atlas University, 34408 Istanbul, Turkey

**Keywords:** rotavirus vaccines, hospital admission, neutrophil–lymphocyte ratio, platelet–lymphocyte ratio, systemic immune inflammatory index

## Abstract

Objective: Rotavirus (RV) is one of the most common and important causes of acute gastroenteritis (AGE) in newborns and children worldwide. The aim of this study was to evaluate the effect of the RV vaccine on the natural history of RV infections using the neutrophil–lymphocyte ratio (NLR), platelet–lymphocyte ratio (PLR), and systemic immune inflammatory index (SII) as hematological indexes, clinical findings, and hospitalization. Method: Children aged 1 month to 5 years who were diagnosed with RV AGE between January 2015 and January 2022 were screened, and 630 patients were included in the study. The SII was calculated by the following formula: neutrophil × platelet/lymphocyte. Results: Fever and hospitalization were significantly higher and breastfeeding was significantly lower in the RV-unvaccinated group than in the RV-vaccinated group. The NLR, PLR, SII, and CRP were significantly higher in the RV-unvaccinated group (*p* < 0.05). The NLR, PLR, and SII were significantly higher both in the non-breastfed group than in the breastfed group and in the hospitalized group than in the not hospitalized group (*p* < 0.05). CRP was not significantly different in either the hospitalization group or the breastfeeding group (*p* > 0.05). SII and PLR were significantly lower in the RV-vaccinated group than in the RV-unvaccinated group in both the breastfed and non-breastfed subgroups. For NLR and CRP, while there was no significant difference according to RV vaccination status in the breastfed group, there was a significant difference in the non-breastfed group (*p* value: <0.001; <0.001). Conclusions: Despite the low level of vaccine coverage, the introduction of RV vaccination had a positive impact on the incidence of RV-positive AGE and related hospitalizations in children. These results showed that breastfed and vaccinated children were less prone to inflammation because their NLR, PLR, and SII ratios were lower. The vaccine does not prevent the disease 100%. However, it can prevent severe disease with exsiccation or death.

## 1. Introduction

Rotaviruses (RVs) are nonenveloped viruses with a segmented double-stranded RNA genome consisting of 11 segments with a total size of 18.55 kilo base pairs in the family Reoviridae. Their genomes encode six structural proteins (VP1–VP6) and six nonstructural proteins (NSP1–NSP6). The genus RV has twelve species (A, B, C, D, E, F, G, H, I, J, K, and L), and RVA is the main human pathogen, causing 90% of diseases. RVA is classified into two genotypes, G and P, and approximately 41 G and 57 P types have been reported [1,2,3,4]. Acute gastroenteritis (AGE) is an infectious disease accompanied by nausea, vomiting, diarrhea, and abdominal pain. Enteropathogenic viruses are the main pathogens of AGE in children worldwide. RVs are the main cause of pediatric AGE among enteropathogenic viruses. RVA causes more than 95% of RVGE [1,5].

Vaccination is an important and effective strategy for the prevention and control of RVA infection [6]. To date, few RV vaccines (Rotarix, RotaTeg, Rotovac, ROTASIIL, and Rotavin) have been preapproved by the World Health Organization (WHO). Due to the use of RV vaccines, reductions in diarrhea and hospital admissions have been observed in many countries. The criterion for successful vaccination is high coverage of the target population, and the higher the vaccine coverage, the greater the significant reductions in hospitalization rates for RVA [7,8,9]. RV vaccines do not fully protect young children against infection, but it does reduce the severity of the infection. In developed countries, it is highly effective in preventing severe AGE, especially in children under 5 years of age [10]. Whereas vaccine effectiveness was high in high-income countries with protection rates against severe RV disease at 80–90%, it was 30–50% lower in low- and middle-income countries [11,12]

The most commonly used indicator of inflammation in laboratory findings is the increase in erythrocyte sedimentation rate (ESH) and C-reactive protein (CRP) levels. Complete blood count (CBC) is simple and inexpensive but contains important follow-up parameters for many diseases. The neutrophil–lymphocyte ratio (NLR) and platelet–lymphocyte ratio (PLR) are inexpensive and easily calculable indexes that correlate with the prognosis of systemic inflammatory diseases [13]. A new marker of inflammation, called the systemic immune inflammatory index (SII), in which all these parameters are used together, was used for the first time in hepatocellular carcinomas in 2014. This marker, obtained by multiplying the neutrophil and platelet counts and dividing the calculation product by the lymphocyte count, has been used to determine inflammation in all research areas, especially in oncology [14].

To the best of our knowledge, however, there are no studies regarding inflammation markers in predicting the effect of vaccines on inflammation markers in children with RV. The aim of this study was to evaluate the effect of the RV vaccine on the natural history of rotavirus infections with the NLR, PLR, and SII as hematological indexes, clinical findings, and hospitalization. To evaluate inflammation, inexpensive, noninvasive, new generation inflammation markers with parameters obtained from routine whole blood examination of each patient were used in this study.

## 2. Materials and Methods

### 2.1. Study Design and Participants

Ethical approval of this study was obtained by the Non-Interventional Ethics Committee of the Medical Faculty of Istanbul Atlas University (21.12.2021; No:10984). The study was performed in accordance with the Helsinki Declaration, and informed consent was obtained from the families of all patients prior to their inclusion in the study.

For the study, children aged 1 month to 5 years who were diagnosed with RV GE (outpatient or inpatient) in Medicine Hospital between January 2015 and January 2022 were screened, and 630 patients were included in the study. A flow chart of the selection of cases is shown in Figure 1.

Demographic characteristics, clinical features (number of diarrhea, duration of diarrhea, number of vomiting, duration of vomiting, presence of fever, presence of dehydration), and laboratory features (transaminase values, kidney functions, serum electrolytes, CBC, stool microbiological examinations) were recorded from the patient’s medical records.

### 2.2. Inclusion Criteria

Patients with AGE (appearing as an outpatient or while in the hospital), between 1 month and 5 years of age, with positive RV antigen in stool, and whose full records could be accessed were included in the study.

### 2.3. Exclusion Criteria

Patients with cultures other than RV in stool examinations or parasites in direct microscopic examination, liver disease, those using potentially hepatotoxic drugs before RV GE, those with a known chronic disease or a history of chronic drug use, those with immunodeficiency, patients with prolonged diarrhea (patients with diarrhea lasting longer than 14 days), patients in the neonatal period and over 5 years of age, and patients with insufficient file data were excluded from the study. Those with non-gastrointestinal system (GIS) symptoms and those who stayed in the intensive care unit (ICU) were excluded from the study.

### 2.4. Rotavirus Vaccination

Oral monovalent RV vaccine (Rotarix^®^, GlaxoSmithKline (GSK), Biological SA, Rixensart, Belgium) was given at 3 and 5 weeks of age. Each child’s RV vaccination status was categorized as complete (two doses) or unvaccinated. RV vaccines were given to all children by the same doctor in the study. The first dose of RV vaccine was not given if the child exceeded 14 weeks, and the second dose was not given if the child exceeded 6 months.

### 2.5. Laboratory Assessments

Venous blood samples were taken from all patients at the time of admission. For CBC, 0.5–2 mL of blood was drawn into purple capped ethylenediaminetetraacetic acid (EDTA) tubes and measured in an automatic blood count device (Sysmex XN 1000, Roche Diagnostics GmbH, Mannheim, Germany) within 1 h at the latest. The NLR and PLR were calculated as [platelet count x neutrophil count]/lymphocyte count. The SII was calculated as platelet count × neutrophil count/lymphocyte count [13]. Routine biochemical parameters were measured in an autoanalyzer (ARCHITECT c8000 Abbott, Sumner, WA, USA).

The diagnosis of RV AGE was made by the presence of positive antigens in the stool. The presence of RV antigens in fresh stool samples was determined by an immunochromatographic method using a chromatography kit (NADAL, Dresden, Germany).

### 2.6. Statistical Analysis

The Statistical Package for the Social Sciences version 21.0 software package for Windows (IBM Corp., Armonk, NY, USA) and JASP 0.17.1 were used for data evaluation and analysis. Categorical variables are presented as frequencies (*n*) and percentages (%), and numerical variables are presented as median (interquartile range). The Kolmogorov–Smirnov test was applied for normality analysis. The chi-square test was used to compare the distribution of categorical variables between groups. The Mann–Whitney U test was used to compare continuous variables between two independent groups. A value of *p* < 0.05 was accepted as statistically significant.

## 3. Results

Of the 630 participants included, 49.2% (*n*: 310) received RV vaccination. A total of 57.8% of the participants were male. A total of 26.3% of the participants had fever and 29.5% of them were hospitalized. The breastfeeding rate was 24%. Table 1 presents the relationship between RV vaccination status and demographic and clinical characteristics. RV-vaccinated and unvaccinated groups were similar in terms of age and gender (*p*: 0.114; 0.266). Fever was found to be significantly lower in RV-vaccinated than in RV-unvaccinated (*p*: 0.022; 22.3%; 30.3%). Hospitalization was found to be significantly lower in RV-vaccinated than in RV-unvaccinated (*p*: <0.001; 19.7%; 39.1%). In addition, breast milk feeding was significantly higher in RV-vaccinated than in RV-unvaccinated (*p*: <0.001; 31.6%; 16.6%).

Table 2 presents the laboratory findings of the RV-vaccinated and RV-unvaccinated patients with RV infection. Platelet, neutrophil count, ALT, urea, sodium, and potassium were similar in the RV-vaccinated and RV-unvaccinated groups. The WBC count was significantly higher in the RV-unvaccinated group than in the RV-vaccinated (*p*: 0.045). Hemoglobin and hematocrit were significantly higher in the RV-unvaccinated group than in the RV-vaccinated group (*p*: <0.001; <0.001). Lymphocyte and lymphocyte % were significantly higher in the RV-vaccinated group than in the RV-unvaccinated group (*p* < 0.001; <0.001). Although neutrophil counts were similar between the two groups, the neutrophil% was significantly higher in the RV-unvaccinated group. The median NLR was 3.27 (1.53–6.73) in the RV-unvaccinated group and 1.4 (0.98–2.03) in the RV-vaccinated group. The median PLR was 148.99 (81.83–254.09) in the RV-unvaccinated group and 67.73 (48.9–115.93) in the RV-vaccinated group. The NLR and PLR were significantly higher in the RV-unvaccinated group than in the RV-vaccinated group (*p* < 0.001; *p* < 0.001). (Figure 2a,b). The median SII was significantly higher in the RV-unvaccinated group than in the RV-vaccinated group (9678 (3822–19,244); 1952 (968–6314); *p*: <0.001). (Figure 2c). In addition, the median CRP level was significantly higher in the RV-unvaccinated group than in the RV-vaccinated group (6.29 (1.36–21); 2.08 (0.64–9.51); *p* < 0.001).

Table 3 presents the laboratory findings of breastfed and non-breastfed children with RV infections. WBC, platelet, neutrophil count, monocyte count, monocyte %, CRP, ALT, AST, urea, creatinine, sodium, and potassium were similar in the breastfed and non-breastfed groups. Hemoglobin and hematocrit were significantly higher in the non-breastfed group than in the breastfed group (*p* = 0.012; 0.003). Lymphocyte and lymphocyte % were significantly higher in the breastfed group than in the non-breastfed group (*p* = 0.006; 0.007). Although neutrophil counts were similar between the two groups, the neutrophil% was significantly higher in the non-breastfed group. The median NLR was 1.91 (1.09–4.86) in the non-breastfed group and 1.77 (1.18–2.6) in the breastfed group. The median PLR was 110.49 (61.2–212.8) in the non-breastfed group and 80.93 (54.47–148.91) in the breastfed group. The NLR and PLR were significantly higher in the non-breastfed group than in the breastfed group (*p* < 0.001; *p* < 0.001). The median SII was significantly higher in the non-breastfed group than in the breastfed group (5826 (1530–14,842); 3581 (1299–9282); *p* < 0.001).

Subgroup analyses were performed because of differences in clinical presentation according to both breastfeeding and vaccination status (Table 4). For NLR and CRP, while there was no significant difference according to RV vaccination status in the breastfed group (*p* value: 0.479; 0.573), there was a significant difference in the non-breastfed group (*p* value: <0.001; <0.001) (Figure 3a). SII and PLR were significantly lower in the RV-vaccinated group than in the RV-unvaccinated group in both the breastfed and non-breastfed subgroups (SII = *p* value: 0.020; <0.001; PLR = *p* value: 0.047; <0.001). (Figure 3b,c).

Table 5 presents the laboratory findings of hospitalized and not hospitalized children with RV infections. The median WBC, HGB, HCT, neutrophil, neutrophil%, NLR, PLR, and SII were significantly higher in the hospitalized groups than in the not-hospitalized group (*p* < 0.05). (Figure 4a–c). The median PLT, LYM, and LYM% were significantly lower in the hospitalized group than in the not hospitalized group (*p* < 0.05). The median CRP level was not significantly different between the hospitalization groups (*p* = 0.418).

## 4. Discussion

A large percentage (70%) of RV diarrhea in Turkey is seen in children under the age of two [15]. In the current study, RV positivity was found to be significantly lower in those who were breastfed than in those who did not receive breast milk. CRP, WBC, neutrophil levels, PLR, NLR and SII were found to be significantly higher in patients who were not vaccinated with RV than in those who were vaccinated. Breast milk may have a protective effect on RV infection. High PLR, NLR, and SII can be used as biomarkers in the differential diagnosis of RV infection. These results showed that the children with the RV vaccine showed less exposure to inflammation.

The incidence of RV infections, which are responsible for the large majority of childhood GEs, is high in both developing and developed countries [11,12]. Determining the etiological factors in these cases is extremely important in treatment and follow-up. It will also prevent the unnecessary use of antibiotics. According to previous and recent comparative studies conducted around the world, the decrease in the frequency of RV in some developed countries is due to vaccination programs. After the first use of the RV vaccine in the USA and European Union countries in 2006, many studies have been conducted on the effect on the RV clinic. At the end of the two seasons in which the vaccine was administered in the USA, a 2-month deviation was observed in the months when RV was most common [16]. Although the burden of rotavirus has decreased during the past decade, rotavirus continues to be the leading cause of diarrhea-associated mortality among children younger than 5 years, responsible for nearly 130,000 deaths annually [17,18]. According to WHO data, approximately five hundred thousand children die from severe RV infections, which can be prevented by vaccination worldwide. In 2009, the WHO Strategic Advisory Group of Experts recommended the employment of routine RV vaccination in expanded immunization programs for infants worldwide, with a particular emphasis on developing countries with a high disease burden [19,20]. Therefore, routine RV vaccination is strongly recommended for all countries. In the current study, of the 630 participants included, 49.2% had received RV vaccinations (complete vaccination). In Turkey, the vaccination rates have remained low because the RV vaccine, which has been in use since 2007, is not in the routine vaccination calendar and due to its high cost. There are no studies on the effect of RV on the clinic after RV vaccination in Turkey. There is no antiviral agent in the treatment against RV; however, partial protection and therefore a decrease in complications and hospitalization rates is possible with vaccination performed during infancy.

Although breast feeding is generally thought to at least reduce the severity of the disease, its specific role in preventing RV diarrhea is controversial [21,22,23,24,25,26,27,28]. The WHO has recommended continuing breastfeeding during RV diarrhea [29]. It has been reported that the severity and duration of the disease in breastfed children are low [25]. Linhares A et al. [28] reported that children aged 0–3 years in Brazil had no evidence of protection against clinical RV by breastfeeding. Differences in study designs and the age of the populations studied have led to differences in study results regarding the effect of breastfeeding on RV diarrhea. In the current study, all RV groups (24%), vaccinated groups (31.6%), and unvaccinated groups (16.6%) had lower breastfeeding rates. Our study has shown that breastfeeding does not reduce the effectiveness of the vaccine. This can be attributed to the protective effect of breast milk, through antibodies passed from the mother and the immature intestinal epithelium. Since lactoferrin and lactadherin, which have anti-rotaviral properties, and anti-rotaviral IgA are found in breast milk, breastfeeding infants with RV AGE is beneficial for treatment [30].

In systemic inflammation, changes in peripheral blood, such as neutrophilia, lymphopenia and thrombocytosis, are observed. Recently, the determination of inflammation has been facilitated using new biomarkers that can be easily calculated with whole blood parameters in the determination of systemic inflammation. The NLR is an indicator of systemic inflammation based on CBC values. In general, the blood neutrophil count increases with the progression of inflammatory diseases. Suastika et al. [31] found that the NLR was significantly higher in severe COVID-19 patients than in nonsevere COVID-19 patients (*n* = 411). Zhang et al. [13] found a significant decrease in LMR and an increase in NLR, as there was an increase in NEU and MON and a decrease in LYM levels in children with RV. These changes in the peripheral blood of children with RV can help us improve our understanding of host immune and inflammatory responses against RV infections. Meanwhile, LMR and NLR can be useful markers in the diagnosis and differentiation of RV infection in AGE children in low medical resource environments. Unlike GE caused by bacterial pathogens, RV infections cause nonbloody diarrhea that is relatively short-lived and associated with a limited inflammatory response. Both viral and host factors are effective in the formation and course of the disease [32]. Protection against RV infection, reducing hospitalization in cases of illness, and minimizing mortality are the main goals of RV vaccines [30]. In the current study, the NLR, PLR, SII, and CRP levels were found to be significantly higher in the RV-vaccinated group than in the RV-unvaccinated group. RV vaccines are one of the most important treatment methods developed to reduce the severity of diarrhea, shorten its duration, and prevent diarrhea. In our study, the inflammatory state in the unvaccinated group may reflect the clinical picture that causes inflammation of the stomach and small intestine. Vaccination reduces the severity of inflammation.

RV infections peak at 6–24 months of life. The fact that it is not seen frequently in babies in the first 3 months can be attributed to the protective effect of breast milk, through the antibodies passed from the mother and the immature intestinal epithelium. The fact that the first infection provides protection for successive infections causes RV infections to be seen rarely or asymptomatically in children over 5 years old [33,34]. In the current study, NLR, PLR, and SII levels were found to be significantly higher in the non-breastfed group than in the breastfed group. Of greatest concern is the difference in immune markers between the breast milk and formula groups. This may be influenced by breast milk rather than vaccines. SII and PLR were significantly lower in the RV-vaccinated group than in the RV-unvaccinated group in both the breastfed and non-breastfed subgroups. There was no significant difference between the NLR, PLR, and SII between those who were breastfed and not vaccinated and those who were not breastfed. It has been shown that breast milk does not reduce the effectiveness of the vaccine [35]. Various studies have shown that breast milk contains antibodies against RV. Its mechanism of action is to inhibit the replication ability of this virus. There is significant benefit in the prevention of RV diarrhea among children by practicing exclusive breastfeeding throughout the first 6 months of life. Therefore, this study provides further evidence for promoting exclusive breastfeeding practices among mothers [36]. Our study findings revealed that breastfeeding is protective against RV inflammation in children.

Fischer et al. [37] reported that 27% of RV-related hospitalizations in developed countries and 32% in developing countries were due to nosocomial infections. The occurrence of significant reductions in hospitalizations and deaths, as well as decreases in the proportion of diarrhea episodes due to RV was observed among the countries that adopted universal RV vaccination [38]. In the current study, NLR, PLR, and SII levels were found to be significantly higher in the hospitalized group than in the nonhospitalized group. The hospitalization rate of the vaccinated (19.7%) was also found to be significantly higher than the nonvaccinated hospitalization rate (39.1%). This inflammatory condition in our study may reflect the clinical picture that causes stomach and small intestine inflammation. However, the most effective way to prevent RV infection is the RV vaccine. Even though the RV vaccine is not 100% effective, according to the results of our study, the rates of hospitalization due to inflammation and complications caused by RV have decreased. The biennial peak in RV and AGE hospitalizations in the USA since vaccine introduction could be driven in part by incomplete vaccination and therefore the build-up of susceptible children in consecutive birth cohorts [39]. In our study, the children who were vaccinated were fully vaccinated (two doses).

As the vaccine has no serious side effects, lower fever, short duration of diarrhea, less severe, shorter hospital stays, and lower levels of inflammatory markers may be due to vaccination of children and breastfeeding. We think that lymphocyte, NLR, PLR, and SII ratios can lead to diagnoses in such cases, especially in regions with laboratory insufficiency, to predict clinical severity and allow an early and cost-effective treatment decision to be made if they are compatible with clinical symptoms. Since the data on the subject are limited, especially in Turkey, there is a need for comprehensive studies with more detailed analyses in larger populations. In conclusion, new inflammatory indexes used as biomarkers to assess the severity of infections can also be used as biomarkers to assess disease severity in RV patients. New-generation inflammation markers, which are simple, inexpensive, easily accessible and noninvasive, obtained using whole blood parameters, can be used as biomarkers in the differential diagnosis of RV infections.

## Figures and Tables

**Figure 1 vaccines-11-00580-f001:**
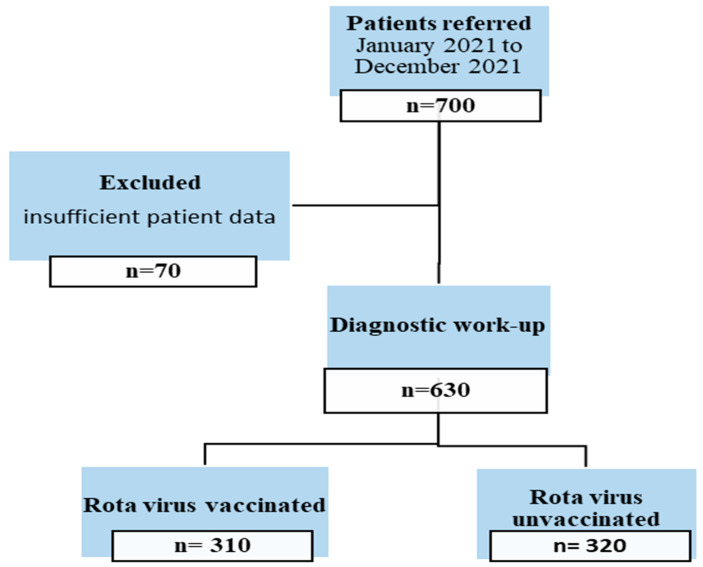
A flow chart of the selection of cases.

**Figure 2 vaccines-11-00580-f002:**
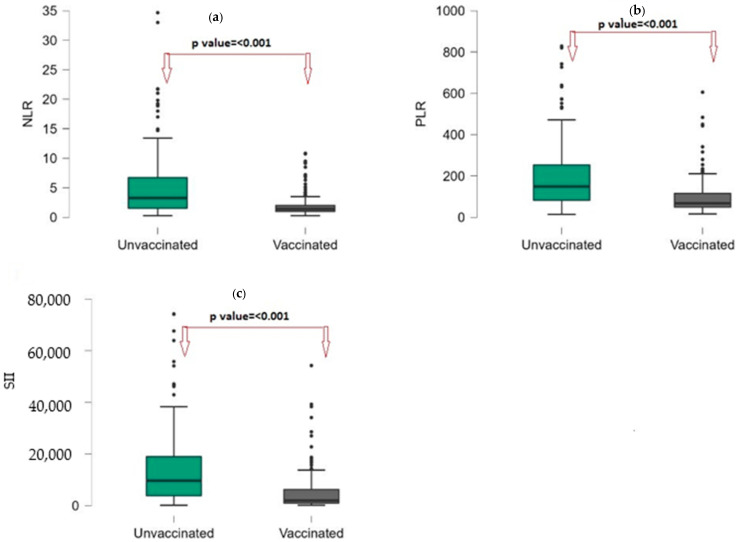
Box-plot graphs by vaccination groups, (**a**): NLR; (**b**): PLR; (**c**): SII. Points(●) are represents the outliers.

**Figure 3 vaccines-11-00580-f003:**
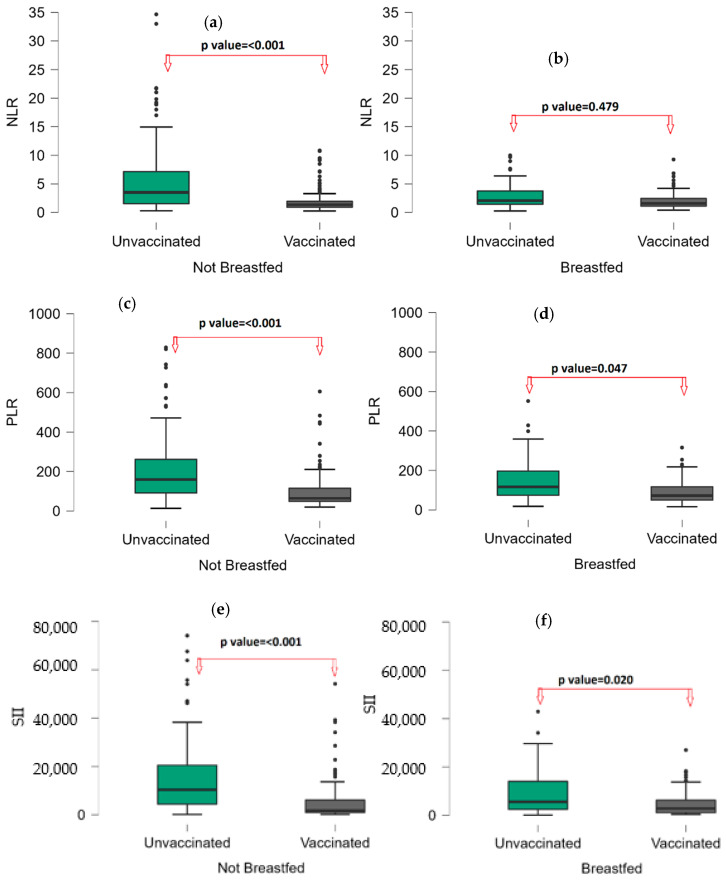
Box-plot graph by vaccination group in breastfeeding subgroups, (**a**): NLR in not breastfed; (**b**): NLR in breastfed; (**c**): PLR in not breastfed; (**d**): PLR in breastfed; (**e**): SII in not breastfed; (**f**): SII in breastfed; Points (●) are represents the outliers.

**Figure 4 vaccines-11-00580-f004:**
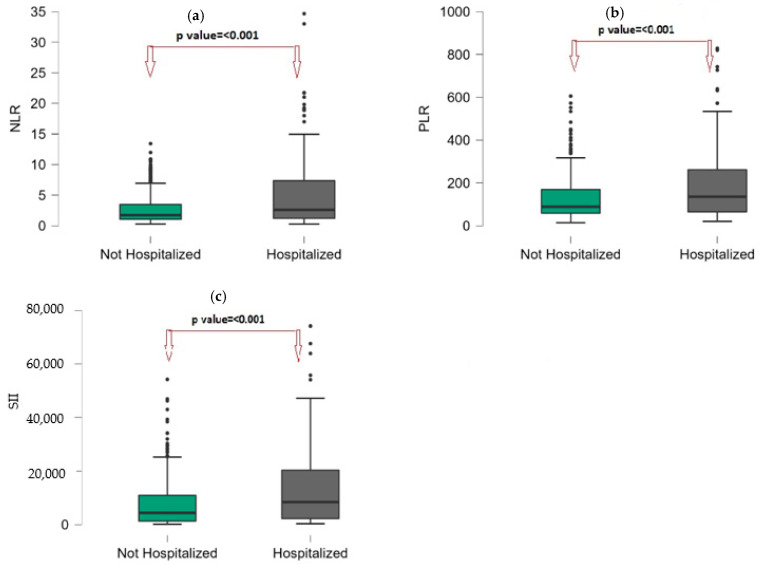
Box-plot graph by hospitalization groups, (**a**): NLR; (**b**): PLR; (**c**): SII. Points (●) are represents the outliers.

**Table 1 vaccines-11-00580-t001:** Demographic and clinical characteristics of children by rotavirus vaccination status.

Characteristics	All Group	Rotavirus Vaccinated	Rotavirus Unvaccinated	*p* Value
	(*n*: 630; %)	(*n*: 310; 49.2%)	(*n*: 320; 50.8%)	
Age (month)	27.5 (15–46)	29 (15–52)	27 (15–43)	0.114 ‡
Gender				0.266 *
Male	364 (57.8)	186 (60)	178 (55.6)	
Female	266 (42.2)	124 (40)	142 (44.4)	
Fever				0.022 *
No	464 (73.7)	241 (77.7)	223 (69.7)	
Yes	166 (26.3)	69 (22.3)	97 (30.3)	
Hospitalized				<0.001 *
No	444 (70.5)	249 (80.3)	195 (60.9)	
Yes	186 (29.5)	61 (19.7)	125 (39.1)	
Breastfed				<0.001 *
No	479 (76)	212 (68.4)	267 (83.4)	
Yes	151 (24)	98 (31.6)	53 (16.6)	

‡: Mann–Whitney U test; * Chi-square test was applied.

**Table 2 vaccines-11-00580-t002:** Laboratory findings of vaccinated and unvaccinated patients with rotavirus infection.

	All Group (*n* = 630)	Rotavirus Vaccinated (*n* = 310)	Rotavirus Unvaccinated (*n* = 320)	*p* Value ‡
WBC (10^3^/µL)	10,030 (8048–12,533)	9885 (7598–12,133)	10,175 (8193–13,160)	0.045
HGB (g/dL)	11.8 (11.1–12.4)	11.5 (11–12.13)	12.1 (11.3–12.7)	<0.001
HCT (%)	35.3 (33.1–37.5)	34.1 (32.38–36.2)	36.5 (34.1–37.8)	<0.001
PLT (10^3^/mL)	321 (261–378)	320 (265–387)	322 (258–371)	0.542
LYM (10^3^/µL)	3.34 (1.56–5.62)	4.97 (2.45–6.61)	2 (1.25–3.9)	<0.001
LYM %	34.55 (17.15–60.75)	54.9 (32.55–67.73)	21.4 (12.1–36.88)	<0.001
Neutrophil (10^3^/µL)	6.57 (4.28–8.93)	7.13 (4.11–8.73)	6.49 (4.49–9.32)	0.261
Neutrophil %	52.25 (25.75–71.8)	31.3 (19.7–54.4)	65.6 (45.925–77.75)	<0.001
Monocyte (10^3^/µL)	0.9 (0.68–1.18)	0.82 (0.64–1.09)	0.98 (0.7–1.3)	<0.001
Monocyte (%)	9 (6.9–12.4)	8.5 (6.55–11.9)	9.4 (7.425–12.8)	0.001
NLR	1.84 (1.1–4.2)	1.4 (0.98–2.03)	3.27 (1.53–6.73)	<0.001
PLR	98.89 (59.4–196.48)	67.73 (48.9–115.93)	148.99 (81.83–254.09)	<0.001
SII	4430 (1817–9114)	1952 (968–6314)	9678 (3822–19,244)	<0.001
CRP (mg/L)	4.31 (0.8–12.3)	2.08 (0.64–9.51)	6.29 (1.36–21)	<0.001
ALT (U/L)	20.3 (15.4–25)	19 (15.4–26)	21 (16.08–24)	0.469
AST (U/L)	37 (29.78–42.93)	36 (29–40.3)	38.2 (32–43)	0.028
Urea (mg/dL)	10.7 (7.08–15.7)	9.5 (6.54–16.6)	11.21 (7.85–14.9)	0.155
Creatinine (mg/dL)	0.47 (0.4–0.63)	0.46 (0.37–0.55)	0.52 (0.43–0.75)	<0.001
Sodium (mEq/L)	137 (135–139)	137 (136–139)	137 (134–139)	0.286
Potassium (mEq/L)	4.2 (3.87–4.6)	4.2 (3.62–4.64)	4.235 (3.91–4.59)	0.06

Abbreviations: WBC: White blood cell; HGB: hemoglobin; HCT: hematocrit; PLT: platelet; LYM: lymphocyte; LYM (%): percentage of lymphocytes; CRP: C reactive protein; ALT: alanine aminotransferase; AST: aspartate aminotransferase. ‡: Mann–Whitney U test was applied.

**Table 3 vaccines-11-00580-t003:** Laboratory findings of breastfed and non-breastfed children with rotavirus infections.

	All Group (*n* = 630)	Breastfed (*n* = 151)	Non-Breastfed (*n* = 479)	*p* Value ‡
WBC (10^3^/µL)	10,030 (8048–12,533)	9880 (7460–12,010)	10,120 (8100–12,710)	0.309
HGB (g/dL)	11.8 (11.1–12.4)	11.5 (10.9–12.2)	11.8 (11.1–12.5)	0.012
HCT (%)	35.3 (33.1–37.5)	34.6 (32.6–36.9)	35.5 (33.3–37.6)	0.003
PLT (10^3^/mL)	321 (261–378)	319 (255–384)	322 (263–376)	0.537
LYM (10^3^/µL)	3.34 (1.56–5.62)	3.76 (2.15–5.82)	3 (1.44–5.61)	0.006
LYM (%)	34.55 (17.15–60.75)	38.6 (21.7–61.9)	34 (14.1–59.6)	0.007
Neutrophil (10^3^/µL)	6.57 (4.28–8.93)	6.66 (4.14–8.87)	6.52 (4.3–8.97)	0.734
Neutrophil (%)	52.25 (25.75–71.8)	45.9 (24–64.4)	53.3 (26.9–74.2)	0.003
Monocyte (10^3^/µL)	0.9 (0.68–1.18)	0.88 (0.69–1.19)	0.9 (0.67–1.16)	0.798
Monocyte (%)	9 (6.9–12.4)	9.3 (7.7–12.6)	8.9 (6.8–12.3)	0.106
NLR	1.84 (1.1–4.2)	1.77 (1.18–2.6)	1.91 (1.09–4.86)	0.029
PLR	98.89 (59.4–196.48)	80.93 (54.47–148.91)	110.49 (61.2–212.8)	0.001
SII	4430 (1817–9114)	3581 (1299–9282)	5826 (1530–14,842)	0.001
CRP (mg/L)	4.31 (0.8–12.3)	5.54 (0.86–12.66)	4.06 (0.79–12.3)	0.194
ALT (U/L)	20.3 (15.4–25)	20 (15.4–25)	20.4 (15.4–25)	0.643
AST (U/L)	37 (29.78–42.93)	36 (29.1–40.3)	37 (30–43)	0.588
Urea (mg/dL)	10.7 (7.08–15.7)	10.7 (6.9–16.7)	10.79 (7.24–15.6)	0.888
Creatinine (mg/dL)	0.47 (0.4–0.63)	0.47 (0.4–0.62)	0.47 (0.4–0.63)	0.73
Sodium (mEq/L)	137 (135–139)	137 (136–139)	137 (135–139)	0.13
Potassium (mEq/L)	4.2 (3.87–4.6)	4.2 (3.87–4.64)	4.2 (3.86–4.6)	0.527

Abbreviations: WBC: White blood cell; HGB: hemoglobin; HCT: hematocrit; PLT: platelet; LYM: lymphocyte; LYM (%): percentage of lymphocytes; CRP: C reactive protein; ALT: alanine aminotransferase; AST: aspartate aminotransferase. ‡: Mann–Whitney U test was applied.

**Table 4 vaccines-11-00580-t004:** Laboratory findings of vaccinated and unvaccinated patients with rotavirus infection in breastfed subgroups.

	Breastfed			Non-Breastfed		
	Rotavirus Vaccinated	Rotavirus Unvaccinated	*p* Value	Rotavirus Vaccinated	Rotavirus Unvaccinated	*p* Value ‡
WBC (10^3^/µL)	8520 (6800–10,373)	8900 (7460–11,553)	0.135	9840 (7670–12,140)	10,250 (8200–13,270)	0.054
HGB (g/dL)	11.8 (11.1–12.3)	11.8 (11–12.6)	0.966	11.5 (11–12.2)	12.1 (11.4–12.8)	<0.001
HCT (%)	35.2 (32.8–37.8)	35.4 (33.1–37.5)	0.856	34.4 (32.4–37)	36.8 (34.2–38)	<0.001
PLT (10^3^/mL)	317 (273–362)	301 (251–359)	0.104	326 (265–388)	322 (260–368)	0.415
LYM (10^3^/µL)	4.37 (3–5.74)	3.8 (2.7–5.19)	0.005	4.98 (2.46–6.73)	1.93 (1.17–3.66)	<0.001
LYM (%)	33 (20–57)	28 (15–47)	0.011	56 (31.6–69.15)	20.3 (10.9–35)	<0.001
Neutrophil (10^3^/µL)	5.28 (3.58–7.97)	4.71 (3.62–7.02)	0.256	6.69 (4.11–8.68)	6.68 (4.5–9.8)	0.077
Neutrophil (%)	51 (31–64)	55 (38–66)	0.025	30 (20–54)	68 (49.3–80.3)	<0.001
Monocyte (10^3^/µL)	0.89 (0.7–1.19)	1 (0.68–1.23)	0.270	0.82 (0.6–1.11)	0.96 (0.71–1.22)	0.001
Monocyte (%)	9.05 (7.37–12.5)	10.45 (8.08–13.8)	0.005	8.3 (6.2–11.95)	9.1 (7.1–12.5)	0.005
NLR	1.36 (0.79–1.91)	1.35 (0.77–2.1)	0.479	1.38 (0.93–2.11)	3.49 (1.6–6.95)	<0.001
PLR	74 (53–98)	80 (61–116)	0.047	66 (50–117)	158 (88–261.67)	<0.001
SII	3438 (1731–6062)	4308 (2504–6407)	0.020	1898 (972–6371)	10,280 (4384–20,460)	<0.001
CRP (mg/L)	1.45 (0.69–3.02)	1.49 (0.76–2.86)	0.573	1.64 (0.64–6.67)	6 (1.27–18.1)	<0.001
ALT (U/L)	20 (15.4–25)	21 (15.4–25)	0.492	19 (15.4–28)	20.9 (16–24)	0.959
AST (U/L)	36 (29.1–41.63)	37 (32–47)	0.148	36 (29–43.2)	38.2 (32–43)	0.152
Urea (mg/dL)	9.5 (6.6–16.6)	10.55 (7.3–15.15)	0.569	9.5 (6.54–16.6)	11.21 (7.85–15)	0.062
Creatinine (mg/dL)	0.47 (0.37–0.56)	0.51 (0.44–0.7)	0.001	0.46 (0.37–0.55)	0.49 (0.42–0.74)	<0.001
Sodium (mEq/L)	137 (136–139)	137 (136–139)	0.916	137 (136–139)	137 (134–139)	0.652
Potassium (mEq/L)	4.2 (3.7–4.6)	4.3 (3.9–4.7)	0.023	4.2 (3.6–4.6)	4.2 (3.9–4.6)	0.111

Abbreviations: WBC: White blood cell; HGB: hemoglobin; HCT: hematocrit; PLT: platelet; LYM: lymphocyte; LYM (%): percentage of lymphocytes; CRP: C reactive protein; ALT: alanine aminotransferase; AST: aspartate aminotransferase. ‡: Mann–Whitney U test was applied.

**Table 5 vaccines-11-00580-t005:** Laboratory findings of hospitalized and not hospitalized children with rotavirus infection.

	All Group (*n* = 630)	Not Hospitalized (*n*: 444; 70.5%)	Hospitalized (*n*: 186; 29.5%)	*p* Value ‡
WBC (10^3^/µL)	10,030 (8048–12,533)	9880 (7855–12,030)	10,545 (8170–14,000)	0.013
HGB (g/dL)	11.8 (11.1–12.4)	11.7 (11.1–12.3)	12 (11.2–12.7)	0.005
HCT (%)	35.3 (33.1–37.5)	35.1 (32.9–37.2)	35.95 (33.5–37.8)	0.013
PLT (10^3^/mL)	321 (261–378)	328 (265–385)	307 (247–357)	0.005
LYM (10^3^/µL)	3.34 (1.56–5.62)	3.53 (1.89–5.92)	2.29 (1.17–5.12)	<0.001
LYM %	34.55 (17.15–60.75)	37.25 (19.6–61.4)	25.3 (11.2–52.1)	<0.001
Neutrophil (10^3^/µL)	6.57 (4.28–8.93)	6.26 (4.14–8.725)	7.695 (4.53–9.8)	0.016
Neutrophil %	52.25 (25.75–71.8)	47.6 (24.9–68.85)	61.7 (34.4–80.4)	<0.001
Monocyte (10^3^/µL)	0.9 (0.68–1.18)	0.9 (0.68–1.195)	0.855 (0.67–1.15)	0.565
Monocyte (%)	9 (6.9–12.4)	9.2 (7.1–12.5)	8.6 (6.7–11.9)	0.07
NLR	1.84 (1.1–4.2)	1.73 (1.07–3.49)	2.61 (1.19–7.41)	<0.001
PLR	98.89 (59.4–196.48)	88.79 (58.65–169.3)	135.59 (64.7–261.87)	<0.001
SII	4430 (1817–9114)	4384 (1321–10,959)	8375 (2215–20,460)	<0.001
CRP (mg/L)	4.31 (0.8–12.3)	4.21 (0.8–11.45)	4.65 (0.78–15)	0.418
ALT (U/L)	20.3 (15.4–25)	20.4 (15.4–25.5)	20 (15.4–23)	0.244
AST (U/L)	37 (29.78–42.93)	36.1 (29.1–42)	38.2 (32–43.2)	0.118
Urea (mg/dL)	10.7 (7.08–15.7)	10.89 (6.9–16.6)	10.2 (7.5–14.77)	0.348
Creatinine (mg/dL)	0.47 (0.4–0.63)	0.47 (0.4–0.59)	0.49 (0.42–0.71)	0.256
Sodium (mEq/L)	137 (135–139)	137 (136–139)	136 (134–138)	0.001
Potassium (mEq/L)	4.2 (3.87–4.6)	4.3 (3.9–4.64)	4.12 (3.76–4.5)	0.021

Abbreviations: WBC: White blood cell; HGB: hemoglobin; HCT: hematocrit; PLT: platelet; LYM: lymphocyte; LYM%: percentage of lymphocytes; CRP: C reactive protein; ALT: alanine aminotransferase; AST: aspartate aminotransferase. ‡: Mann–Whitney U test was applied.

## Data Availability

Participant-level data are available from the corresponding author.

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
