# Peer review of "The Effect of Virus-Specific Vaccination on Laboratory Infection Markers of Children with Acute Rotavirus-Associated Acute Gastroenteritis"

_vaccines, 2023, doi:10.3390/vaccines11030580_

Round 1
Reviewer 1 Report
Dear authors
Thank you for the manuscript I find very well written and M&M well designed. The subject of vaccination against gastroenteritis due to rotavirus is a very interesting subject. It is a disease which can be prevented by vaccination or at least which can be eradicated if the vaccine covers all nations at its maximum. Unfortunately not all countries vaccinate and as a result our young children continue to suffer from acute gastroenteritis.
Through reading your manuscript, I do not find enough arguments to propose markers of predictable inflammation for a rotavirus infection, even if you recommend a diagnosis based on the clinic. Inflammation is a physiological phenomenon that can be associated with both viral and bacterial infections. On the other hand, the inflammation due to rotaviruses cannot distinguished from norovirus, Aichi virus and other enteric viruses which affect this category of the population. It would have been very good to push your diagnosis on the serotypes which circulate in both vaccinated and non-vaccinated children by molecular biology techniques. You have a very good sample pool and in my sense, this will bring the novety in your study.
Author Response
Dear Editor,
First, we would like thank the reviewers for the helpful comments, which led us to conduct appropriate experiments. The manuscript has subsequently been rewritten to address these concerns and comments of the reviewers.
We are grateful for your understanding and cooperation in this matter.
We believe that the manuscript is now suitable for review. We look forward to your reply.
RESPONSE TO REVIEWERS:
Reviewer 1
Open Review
(x) I would not like to sign my review report
( ) I would like to sign my review report
English language and style
( ) English very difficult to understand/incomprehensible
( ) Extensive editing of English language and style required
( ) Moderate English changes required
( ) English language and style are fine/minor spell check required
(x) I don't feel qualified to judge about the English language and style
Yes Can be improved Must be improved Not applicable
Does the introduction provide sufficient background and include all relevant references?
(x) ( ) ( ) ( )
Are all the cited references relevant to the research?
(x) ( ) ( ) ( )
Is the research design appropriate?
( ) ( ) (x) ( )
Are the methods adequately described?
(x) ( ) ( ) ( )
Are the results clearly presented?
(x) ( ) ( ) ( )
Are the conclusions supported by the results?
(x) ( ) ( ) ( )
Comments and Suggestions for Authors
Dear authors
Thank you for the manuscript I find very well written and M&M well designed. The subject of vaccination against gastroenteritis due to rotavirus is a very interesting subject. It is a disease which can be prevented by vaccination or at least which can be eradicated if the vaccine covers all nations at its maximum. Unfortunately not all countries vaccinate and as a result our young children continue to suffer from acute gastroenteritis.
Through reading your manuscript, I do not find enough arguments to propose markers of predictable inflammation for a rotavirus infection, even if you recommend a diagnosis based on the clinic. Inflammation is a physiological phenomenon that can be associated with both viral and bacterial infections. On the other hand, the inflammation due to rotaviruses cannot distinguished from norovirus, Aichi virus and other enteric viruses which affect this category of the population. It would have been very good to push your diagnosis on the serotypes which circulate in both vaccinated and non-vaccinated children by molecular biology techniques. You have a very good sample pool and in my sense, this will bring the novety in your study.
RESPONSE
Thank you for your good comments.
In our next study, we are planning a research project to diagnose circulating serotypes in both vaccinated and unvaccinated children using molecular biology techniques.
Reviewer 2 Report
The work entitled "The Effect of Oral Rotavirus Vaccine on New-Generation In- 1 flammation Markers in Children with Rotavirus Infection" byOkuyan et al. is very well conducted methodologically. The results are robust and the analysis are very relevants.
However, it is necessary to improve the overall writing of the article, make it more fluid, easier to read for the reader. Furthermore, the data presented in the tables are very good, but it is difficult to interpret, so including some type of graph that summarizes that information would greatly enhance the article.
Author Response
Dear Editor,
First, we would like thank the reviewers for the helpful comments, which led us to conduct appropriate experiments. The manuscript has subsequently been rewritten to address these concerns and comments of the reviewers.
We are grateful for your understanding and cooperation in this matter.
We believe that the manuscript is now suitable for review. We look forward to your reply.
RESPONSE TO REVIEWERS:
Reviewer 2
Open Review
( ) I would not like to sign my review report
(x) I would like to sign my review report
English language and style
( ) English very difficult to understand/incomprehensible
( ) Extensive editing of English language and style required
(x) Moderate English changes required
( ) English language and style are fine/minor spell check required
( ) I don't feel qualified to judge about the English language and style
Yes Can be improved Must be improved Not applicable
Does the introduction provide sufficient background and include all relevant references?
(x) ( ) ( ) ( )
Are all the cited references relevant to the research?
(x) ( ) ( ) ( )
Is the research design appropriate?
(x) ( ) ( ) ( )
Are the methods adequately described?
(x) ( ) ( ) ( )
Are the results clearly presented?
( ) (x) ( ) ( )
Are the conclusions supported by the results?
(x) ( ) ( ) ( )
Comments and Suggestions for Authors
The work entitled "The Effect of Oral Rotavirus Vaccine on New-Generation Inflammation Markers in Children with Rotavirus Infection" byOkuyan et al. is very well conducted methodologically. The results are robust and the analysis are very relevants.
However, it is necessary to improve the overall writing of the article, make it more fluid, easier to read for the reader. Furthermore, the data presented in the tables are very good, but it is difficult to interpret, so including some type of graph that summarizes that information would greatly enhance the article.
RESPONSE:
Thank you for this suggestion. As suggested by the reviewer, we have added figures for to pointing main results.
Reviewer 3 Report
RSV infection in babies/children are very important. The symptoms and clinical read outs are known. The value of vaccinations altering symptoms and clinical read outs are also known. The combination of breast feeding as the second immune parameter for the prevention or improving the disease parameters are important and know as well. The data do not allow to find explanations why the descriptions of the clinics are improved. There is no information on antibodies or T cells the immunized children show after vaccination. There is now information on the quality of the breast milk - in particular the content of IgA. There is no objective information as to why there is an improvement for breast fed and or vaccinated children. What is really the novelty of this publication? There is no specific immune parameter to explain the data. Is this not important for the authors?
Author Response
Dear Editor,
First, we would like thank the reviewers for the helpful comments, which led us to conduct appropriate experiments. The manuscript has subsequently been rewritten to address these concerns and comments of the reviewers.
We are grateful for your understanding and cooperation in this matter.
We believe that the manuscript is now suitable for review. We look forward to your reply.
RESPONSE TO REVIEWERS:
Reviewer 3
Open Review
( ) I would not like to sign my review report
(x) I would like to sign my review report
English language and style
( ) English very difficult to understand/incomprehensible
( ) Extensive editing of English language and style required
( ) Moderate English changes required
(x) English language and style are fine/minor spell check required
( ) I don't feel qualified to judge about the English language and style
Yes Can be improved Must be improved Not applicable
Does the introduction provide sufficient background and include all relevant references?
( ) ( ) (x) ( )
Are all the cited references relevant to the research?
( ) (x) ( ) ( )
Is the research design appropriate?
( ) ( ) (x) ( )
Are the methods adequately described?
(x) ( ) ( ) ( )
Are the results clearly presented?
( ) (x) ( ) ( )
Are the conclusions supported by the results?
( ) ( ) ( ) (x)
Comments and Suggestions for Authors
RSV infection in babies/children are very important. The symptoms and clinical read outs are known. The value of vaccinations altering symptoms and clinical read outs are also known. The combination of breast feeding as the second immune parameter for the prevention or improving the disease parameters are important and know as well. The data do not allow to find explanations why the descriptions of the clinics are improved. There is no information on antibodies or T cells the immunized children show after vaccination. There is now information on the quality of the breast milk - in particular the content of IgA. There is no objective information as to why there is an improvement for breast fed and or vaccinated children. What is really the novelty of this publication? There is no specific immune parameter to explain the data. Is this not important for the authors?
RESPONSE
We agree with your valuable opinion but it is not routine to check for antibodies after rota vaccine. Our aim is to examine the markers of inflammation after rota vaccination, and we examined the data in this direction. In addition, because the study was retrospective, antibodies and T cell groups could not be examined. However, the data in the study support our aim.
Maternal factors affecting rotavirus vaccine efficacy have become a focus of attention with research showing that in addition to immunologic components of breast milk (immunoglobulin A & B, (IgA and IgG)), non-immunologic components may also play a role in reduced vaccine efficacy. Very little is known about the association of breast milk components with rotavirus vaccine response.
In our next study, we are planning a research project to diagnose circulating serotypes in both vaccinated and unvaccinated children using molecular biology techniques.
Reviewer 4 Report
The effect of oral rotavirus vaccine on new-generation inflammation markers in children with rotavirus infection
By Omer Okuyan* et al (*Corresponding author)
Submitted to Vaccine(Editorial No. vaccine-2225597)
General Comments
This study explores the relationship of neutrophil-lymphocyte ratio (NLR), platelet-lymphocyte ratio (PLR), a systemic immune inflammatory index (SII) and clinical findings in children with acute gastroenteritis (AGE) due to species A rotavirus (RVA) infection with the presence/absence of previous RV vaccination (Rotarix), breastfeeding Yes/No and hospitalization Yes/No. The investigation was carried out between 2015 and 2022 and comprised 630 children of <5 y of age of whom 310 were RV vaccinated vs 320 who were not. It was found that the NLR, PLR and SII showed significant relationships with previous RV vaccination, with previous breastfeeding and the need for hospitalization whilst the C reactive protein levels were not significantly different. The SII marker is reported to be ‘highly correlated with PLR, NLR and lymphocyte counts’.
The data are interesting. Their significance could be strengthened by more transparency in presentation. The correlations of SII with PLR, NLR and lymphocyte values are considered to be at least partially spurious, since the ratios share common measurements.
Specific Comments
Line
1 Reconsider Title, e.g. ‘The effect of virus-specific vaccination on laboratory infection markers of children with acute rotavirus-associated acute gastroenteritis’, or similar.
16 Read: Rotavirus (RV) is one of the…
18 … history of RV infection using…
19 The SII index should be defined in Abstract.
22 … breastfeeding…
26 … not significantly different…
34 … were less prone to…
35 Omit sentence.
36 … prevent severe disease with exsiccation or death.
42 … consisting of 11 segments of a total size of 18.55 kilo base pairs.
44 There are now 2 additional species (K, L):
Johne R, Schilling-Loeffler K, Ulrich RG, Tausch SH. Whole Genome Sequence Analysis of a Prototype Strain of the Novel Putative Rotavirus Species L. Viruses. 2022 Feb 24;14(3):462. doi: 10.3390/v14030462. PMID: 35336869; PMCID: PMC8954357.
Johne R, Tausch SH, Grützke J, Falkenhagen A, Patzina-Mehling C, Beer M, Höper D, Ulrich RG. Distantly Related Rotaviruses in Common Shrews, Germany, 2004-2014. Emerg Infect Dis. 2019 Dec;25(12):2310-2314. doi: 10.3201/eid2512.191225. PMID: 31742508; PMCID: PMC6874240.
46 The numbers of G and P types should be updated, see: www.ICTVonline/virus taxonomy.
48 … RVs are …
52 Refs 3-6 are not cited in text. Read: … Rotarix, RotaTeq, Rotavac, ROTASIIL, Rotavin…
57 Consider citation of:
Bergman H, Henschke N, Hungerford D, Pitan F, Ndwandwe D, Cunliffe N, Soares-Weiser K. Vaccines for preventing rotavirus diarrhoea: vaccines in use. Cochrane Database Syst Rev. 2021 Nov 17;11(11):CD008521. doi: 10.1002/14651858.CD008521.pub6. PMID: 34788488; PMCID: PMC8597890.
Soares-Weiser K, Bergman H, Henschke N, Pitan F, Cunliffe N. Vaccines for preventing rotavirus diarrhoea: vaccines in use. Cochrane Database Syst Rev. 2019 Oct 28;2019(10):CD008521. doi: 10.1002/14651858.CD008521.pub5. Update in: Cochrane Database Syst Rev. 2021 Nov 17;11:CD008521. PMID: 31684685; PMCID: PMC6816010.
59 The difference of efficacy of RV vaccination at population level in countries of different socio-eonomic standard should be mentioned. Consider citation of:
Desselberger U. Differences of Rotavirus Vaccine Effectiveness by Country: Likely Causes and Contributing Factors. Pathogens. 2017 Dec 12;6(4):65. doi: 10.3390/pathogens6040065. PMID: 29231855; PMCID: PMC5750589.
Parker EP, Ramani S, Lopman BA, Church JA, Iturriza-Gómara M, Prendergast AJ, Grassly NC. Causes of impaired oral vaccine efficacy in developing countries. Future Microbiol. 2018 Jan;13(1):97-118. doi: 10.2217/fmb-2017-0128. Epub 2017 Dec 8. PMID: 29218997; PMCID: PMC7026772.
64 … are inexpensive and … calculable indices that correlate with…
67 Consider phrasing: … This marker, obtained by multiplying the neutrophil and platelet counts and dividing the calculation product by the lymphocyte count,
71 … there are no studies regarding inflammation markers…
87 … and 630 patients…
105 … non-GIS symptoms… Spell out at first mentioning.
119 … calculated as [platelet count x neutrophil count]/lymphocyte count…
165 Table 2. Significant p values should be printed in bold.
166 … hematocrit…
192 Table 4. See comment Table 2.
218 … laboratory findings…
218f Table 6 and Figures 2-4. The correlations presented are considered to be at least partially spurious, since the ratios compared contain common measurements. This portion of the text has to be thoroughly reconsidered.
232f Should read: … Figure 2…. Figure 3… Figure 4… [Figure 1 is shown in line 90]
240 This sentence and citation should be moved to the beginning of Introduction.
254 See comment above (Line 57).
260f The data on RV-associated mortality of <5 y old children are out of date. Consider citation of:
Tate JE, Burton AH, Boschi-Pinto C, Parashar UD; World Health Organization–Coordinated Global Rotavirus Surveillance Network. Global, Regional, and National Estimates of Rotavirus Mortality in Children <5 Years of Age, 2000-2013. Clin Infect Dis. 2016 May 1;62 Suppl 2:S96-S105. doi: 10.1093/cid/civ1013. PMID: 27059362.
Troeger C, Khalil IA, Rao PC, Cao S, Blacker BF, Ahmed T, Armah G, Bines JE, Brewer TG, Colombara DV, Kang G, Kirkpatrick BD, Kirkwood CD, Mwenda JM, Parashar UD, Petri WA Jr, Riddle MS, Steele AD, Thompson RL, Walson JL, Sanders JW, Mokdad AH, Murray CJL, Hay SI, Reiner RC Jr. Rotavirus Vaccination and the Global Burden of Rotavirus Diarrhea Among Children Younger Than 5 Years. JAMA Pediatr. 2018 Oct 1;172(10):958-965. doi: 10.1001/jamapediatrics.2018.1960. Erratum in: JAMA Pediatr. 2022 Feb 1;176(2):208. PMID: 30105384; PMCID: PMC6233802.
282 … 16.6%...
285 Provide relevant refs.
Consider citation of: Angel J, Steele AD, Franco MA. Correlates of protection for rotavirus vaccines: Possible alternative trial endpoints, opportunities, and challenges. Hum Vaccin Immunother. 2014;10(12):3659-71. doi: 10.4161/hv.34361. PMID: 25483685; PMCID: PMC4514048.
304 See comment line 285.
321 and line 338. See comment line 218.
329 … provides further evidence for promoting…
342f Omit sentence.
351 Reviewer does not agree with this argument. Testing for the presence of RVA antigen in feces is now routine in virtually all clinical microbiological diagnostic laboratories.
369 to 277. Refs 3-6 are not cited in text.
430 Read: Velázquez…
448 Read: Figure 3
452 Read: Figure 4
Regarding Figures 2-4 see comments above.
Author Response
Dear Editor,
First, we would like thank the reviewers for the helpful comments, which led us to conduct appropriate experiments. The manuscript has subsequently been rewritten to address these concerns and comments of the reviewers.
We are grateful for your understanding and cooperation in this matter.
We believe that the manuscript is now suitable for review. We look forward to your reply.
RESPONSE TO REVIEWERS:
Referee 4
Open Review
( ) I would not like to sign my review report
(x) I would like to sign my review report
English language and style
( ) English very difficult to understand/incomprehensible
( ) Extensive editing of English language and style required
(x) Moderate English changes required
( ) English language and style are fine/minor spell check required
( ) I don't feel qualified to judge about the English language and style
Yes Can be improved Must be improved Not applicable
Does the introduction provide sufficient background and include all relevant references?
( ) (x) ( ) ( )
Are all the cited references relevant to the research?
(x) ( ) ( ) ( )
Is the research design appropriate?
( ) (x) ( ) ( )
Are the methods adequately described?
(x) ( ) ( ) ( )
Are the results clearly presented?
( ) ( ) (x) ( )
Are the conclusions supported by the results?
( ) (x) ( ) ( )
Comments and Suggestions for Authors
The effect of oral rotavirus vaccine on new-generation inflammation markers in children with rotavirus infection
By Omer Okuyan* et al (*Corresponding author)
Submitted to Vaccine(Editorial No. vaccine-2225597)
General Comments
This study explores the relationship of neutrophil-lymphocyte ratio (NLR), platelet-lymphocyte ratio (PLR), a systemic immune inflammatory index (SII) and clinical findings in children with acute gastroenteritis (AGE) due to species A rotavirus (RVA) infection with the presence/absence of previous RV vaccination (Rotarix), breastfeeding Yes/No and hospitalization Yes/No. The investigation was carried out between 2015 and 2022 and comprised 630 children of <5 y of age of whom 310 were RV vaccinated vs 320 who were not. It was found that the NLR, PLR and SII showed significant relationships with previous RV vaccination, with previous breastfeeding and the need for hospitalization whilst the C reactive protein levels were not significantly different. The SII marker is reported to be ‘highly correlated with PLR, NLR and lymphocyte counts’.
The data are interesting. Their significance could be strengthened by more transparency in presentation. The correlations of SII with PLR, NLR and lymphocyte values are considered to be at least partially spurious, since the ratios share common measurements.
Specific Comments
Line 1 Reconsider Title, e.g. ‘The effect of virus-specific vaccination on laboratory infection markers of children with acute rotavirus-associated acute gastroenteritis’, or similar.
RESPONSE
The title was changed to ''The effect of virus-specific vaccination on laboratory infection markers of children with acute rotavirus-associated acute gastroenteritis.''.
16 Read: Rotavirus (RV) is one of the…
18 … history of RV infection using…
19 The SII index should be defined in Abstract.
22 … breastfeeding…
26 … not significantly different…
34 … were less prone to…
35 Omit sentence.
36 … prevent severe disease with exsiccation or death.
42 … consisting of 11 segments of a total size of 18.55 kilo base pairs.
44 There are now 2 additional species (K, L):
Johne R, Schilling-Loeffler K, Ulrich RG, Tausch SH. Whole Genome Sequence Analysis of a Prototype Strain of the Novel Putative Rotavirus Species L. Viruses. 2022 Feb 24;14(3):462. doi: 10.3390/v14030462. PMID: 35336869; PMCID: PMC8954357.
Johne R, Tausch SH, Grützke J, Falkenhagen A, Patzina-Mehling C, Beer M, Höper D, Ulrich RG. Distantly Related Rotaviruses in Common Shrews, Germany, 2004-2014. Emerg Infect Dis. 2019 Dec;25(12):2310-2314. doi: 10.3201/eid2512.191225. PMID: 31742508; PMCID: PMC6874240.
46 The numbers of G and P types should be updated, see: www.ICTVonline/virus taxonomy.
48 … RVs are …
52 Refs 3-6 are not cited in text. Read: … Rotarix, RotaTeq, Rotavac, ROTASIIL, Rotavin…
57 Consider citation of:
Bergman H, Henschke N, Hungerford D, Pitan F, Ndwandwe D, Cunliffe N, Soares-Weiser K. Vaccines for preventing rotavirus diarrhoea: vaccines in use. Cochrane Database Syst Rev. 2021 Nov 17;11(11):CD008521. doi: 10.1002/14651858.CD008521.pub6. PMID: 34788488; PMCID: PMC8597890.
Soares-Weiser K, Bergman H, Henschke N, Pitan F, Cunliffe N. Vaccines for preventing rotavirus diarrhoea: vaccines in use. Cochrane Database Syst Rev. 2019 Oct 28;2019(10):CD008521. doi: 10.1002/14651858.CD008521.pub5. Update in: Cochrane Database Syst Rev. 2021 Nov 17;11:CD008521. PMID: 31684685; PMCID: PMC6816010.
59 The difference of efficacy of RV vaccination at population level in countries of different socio-eonomic standard should be mentioned. Consider citation of:
Desselberger U. Differences of Rotavirus Vaccine Effectiveness by Country: Likely Causes and Contributing Factors. Pathogens. 2017 Dec 12;6(4):65. doi: 10.3390/pathogens6040065. PMID: 29231855; PMCID: PMC5750589.
Parker EP, Ramani S, Lopman BA, Church JA, Iturriza-Gómara M, Prendergast AJ, Grassly NC. Causes of impaired oral vaccine efficacy in developing countries. Future Microbiol. 2018 Jan;13(1):97-118. doi: 10.2217/fmb-2017-0128. Epub 2017 Dec 8. PMID: 29218997; PMCID: PMC7026772.
64 … are inexpensive and … calculable indices that correlate with…
67 Consider phrasing: … This marker, obtained by multiplying the neutrophil and platelet counts and dividing the calculation product by the lymphocyte count,
71 … there are no studies regarding inflammation markers…
87 … and 630 patients…
105 … non-GIS symptoms… Spell out at first mentioning.
119 … calculated as [platelet count x neutrophil count]/lymphocyte count…
165 Table 2. Significant p values should be printed in bold.
166 … hematocrit…
192 Table 4. See comment Table 2.
218 … laboratory findings…
218f Table 6 and Figures 2-4. The correlations presented are considered to be at least partially spurious, since the ratios compared contain common measurements. This portion of the text has to be thoroughly reconsidered.
232f Should read: … Figure 2…. Figure 3… Figure 4… [Figure 1 is shown in line 90]
240 This sentence and citation should be moved to the beginning of Introduction.
254 See comment above (Line 57).
260f The data on RV-associated mortality of <5 y old children are out of date. Consider citation of:
Tate JE, Burton AH, Boschi-Pinto C, Parashar UD; World Health Organization–Coordinated Global Rotavirus Surveillance Network. Global, Regional, and National Estimates of Rotavirus Mortality in Children <5 Years of Age, 2000-2013. Clin Infect Dis. 2016 May 1;62 Suppl 2:S96-S105. doi: 10.1093/cid/civ1013. PMID: 27059362.
Troeger C, Khalil IA, Rao PC, Cao S, Blacker BF, Ahmed T, Armah G, Bines JE, Brewer TG, Colombara DV, Kang G, Kirkpatrick BD, Kirkwood CD, Mwenda JM, Parashar UD, Petri WA Jr, Riddle MS, Steele AD, Thompson RL, Walson JL, Sanders JW, Mokdad AH, Murray CJL, Hay SI, Reiner RC Jr. Rotavirus Vaccination and the Global Burden of Rotavirus Diarrhea Among Children Younger Than 5 Years. JAMA Pediatr. 2018 Oct 1;172(10):958-965. doi: 10.1001/jamapediatrics.2018.1960. Erratum in: JAMA Pediatr. 2022 Feb 1;176(2):208. PMID: 30105384; PMCID: PMC6233802.
282 … 16.6%...
285 Provide relevant refs.
Consider citation of:
Angel J, Steele AD, Franco MA. Correlates of protection for rotavirus vaccines: Possible alternative trial endpoints, opportunities, and challenges. Hum Vaccin Immunother. 2014;10(12):3659-71. doi: 10.4161/hv.34361. PMID: 25483685; PMCID: PMC4514048.
304 See comment line 285.
321 and line 338. See comment line 218.
329 … provides further evidence for promoting…
342f Omit sentence.
351 Reviewer does not agree with this argument. Testing for the presence of RVA antigen in feces is now routine in virtually all clinical microbiological diagnostic laboratories.
''RV antigen may not be studied all the time and everywhere in pediatric cases presenting with GE clinic.'' was removed.
369 to 277. Refs 3-6 are not cited in text.
References have been edited.
430 Read: Velázquez…
448 Read: Figure 3
452 Read: Figure 4
Regarding Figures 2-4 see comments above.
All comments have been replied in accordance with your suggestions.
Thank you for pointing out “ the correlation between NLR, PLR, SII”. The reviewer is correct.
These are highly correlated as they use the same parameters. We are aware of this.
We decided to show the correlation table (Table 6). Because, the NLR correlations were slightly lower in the vaccinated and breastfed groups compared to the others.
However, if the reviewer thinks this is unnecessary, we are ready to remove it.
We removed the figures and replaced them at the suggestion of one of the other reviewers.
All comments have been replied in accordance with your suggestions.
Round 2
Reviewer 1 Report
Dear authors
I remain decided on performing the molecular analyses, this what will add the scientific value of your reserach.
Author Response
Dear Editor,
First, we would like thank the reviewers for the helpful comments, which led us to conduct appropriate experiments. The re- revised manuscript has subsequently been rewritten to address these concerns and comments of the reviewers.
We are grateful for your understanding and cooperation in this matter.
We believe that the manuscript is now suitable for review. We look forward to your reply.
RESPONSE TO REVIEWERS:
Reviewer 1
(x) I would not like to sign my review report
( ) I would like to sign my review report
English language and style
( ) English very difficult to understand/incomprehensible
( ) Extensive editing of English language and style required
( ) Moderate English changes required
( ) English language and style are fine/minor spell check required
(x) I don't feel qualified to judge about the English language and style
Comments and Suggestions for Authors
Dear authors
I remain decided on performing the molecular analyses, this what will add the scientific value of your reserach.
Thank you to the referee for his valuable comment. Of course, molecular analyzes will have very important contributions to understanding the effects of rotavirus on child health. However, this study is a retrospective study. Our aim in this study was to see the effect of rotavirus vaccination and breastfeeding on the natural history of rotavirus infection. We do not have data on molecular analyzes. A different study can be designed for the purpose of molecular analysis. However, we do not have such data at the moment, this study is a retrospective study.
Reviewer 3 Report
The data are too descriptive and not really novel.
We know that vaccination and breastfeeding improves the health of babies. But the authors provide no reason why this is so
Author Response
Dear Editor,
First, we would like thank the reviewers for the helpful comments, which led us to conduct appropriate experiments. The re- revised manuscript has subsequently been rewritten to address these concerns and comments of the reviewers.
We are grateful for your understanding and cooperation in this matter.
We believe that the manuscript is now suitable for review. We look forward to your reply.
RESPONSE TO REVIEWERS:
Reviewer 3
Open Review
( ) I would not like to sign my review report
(x) I would like to sign my review report
English language and style
( ) English very difficult to understand/incomprehensible
( ) Extensive editing of English language and style required
( ) Moderate English changes required
(x) English language and style are fine/minor spell check required
( ) I don't feel qualified to judge about the English language and style
Yes Can be improved Must be improved Not applicable
Does the introduction provide sufficient background and include all relevant references?
( ) ( ) (x) ( )
Are all the cited references relevant to the research?
( ) ( ) (x) ( )
Is the research design appropriate?
( ) ( ) (x) ( )
Are the methods adequately described?
( ) (x) ( ) ( )
Are the results clearly presented?
(x) ( ) ( ) ( )
Are the conclusions supported by the results?
( ) ( ) (x) ( )
Comments and Suggestions for Authors
The data are too descriptive and not really novel.
We know that vaccination and breastfeeding improves the health of babies. But the authors provide no reason why this is so.
Thanks to the referee for his comment. This study is a retrospective study and the available data are limited. We agree with the referee, vaccination and breastfeeding are 2 important factors that improve child health. However, we have data on 630 Rotavirus-infected children. Actually, this is a large database. Examination of this data in terms of biomarkers contributes to the literature.
As the vaccine has no serious side effects, lower fever, short duration of diarrhea and not severe, shorter hospital stay, and lower levels of inflammatory markers may be due to vaccination of children and breastfeeding.
It has been added to the conclusion section.
Reviewer 4 Report
The authors have considered the comments/suggestions of this reviewer very carefully and virtually accepted all of them. The manuscript has significantly improved.
There is still the issue of correlation of SII values with NLR and PLR values which are considered as partially spurious, since there are common data in the ratios compared. It is suggested to rephrase this portion of the text.
Author Response
Dear Editor,
First, we would like thank the reviewers for the helpful comments, which led us to conduct appropriate experiments. The re- revised manuscript has subsequently been rewritten to address these concerns and comments of the reviewers.
We are grateful for your understanding and cooperation in this matter.
We believe that the manuscript is now suitable for review. We look forward to your reply.
RESPONSE TO REVIEWERS:
Referee 4
Open Review
(x) I would not like to sign my review report
( ) I would like to sign my review report
English language and style
( ) English very difficult to understand/incomprehensible
( ) Extensive editing of English language and style required
( ) Moderate English changes required
(x) English language and style are fine/minor spell check required
( ) I don't feel qualified to judge about the English language and style
Yes Can be improved Must be improved Not applicable
Does the introduction provide sufficient background and include all relevant references?
(x) ( ) ( ) ( )
Are all the cited references relevant to the research?
(x) ( ) ( ) ( )
Is the research design appropriate?
( ) (x) ( ) ( )
Are the methods adequately described?
(x) ( ) ( ) ( )
Are the results clearly presented?
(x) ( ) ( ) ( )
Are the conclusions supported by the results?
( ) (x) ( ) ( )
Comments and Suggestions for Authors
The authors have considered the comments/suggestions of this reviewer very carefully and virtually accepted all of them. The manuscript has significantly improved.
There is still the issue of correlation of SII values with NLR and PLR values which are considered as partially spurious, since there are common data in the ratios compared. It is suggested to rephrase this portion of the text.
As expected, correlations were found between these parameters, as there was common data in the ratios compared between SII, NLR, and PLR values. We agree with the referee on this point. The main purpose of our article is to show the effect of these markers on the natural course of the disease. Therefore, we thought it appropriate to remove the table of correlations from the manuscript.